# Mindfulness versus Physical Exercise: Effects of Two Recovery Strategies on Mental Health, Stress and Immunoglobulin A during Lunch Breaks. A Randomized Controlled Trial

**DOI:** 10.3390/ijerph17082839

**Published:** 2020-04-20

**Authors:** Cintia Díaz-Silveira, Carlos-María Alcover, Francisco Burgos, Alberto Marcos, Miguel A. Santed

**Affiliations:** 1Department of Psychology, Faculty of Health Sciences, Universidad Rey Juan Carlos, Avda. Atenas s/n, Alcorcón, 28922 Madrid, Spain; cintia.diazsilveira@urjc.es (C.D.-S.); carlosmaria.alcover@urjc.es (C.-M.A.); 2Faculty of Psychology, Universidad Nacional de Educación a Distancia, c/Juan del Rosal 10, 29040 Madrid, Spain; psico.fburgos@gmail.com (F.B.); albmarcos@pas.uned.es (A.M.)

**Keywords:** mindfulness meditation, physical exercise, mental health, stress, saliva, immunoglobulin A, recovery, lunch break

## Abstract

This research analyses the effects of mindfulness meditation (MM) and physical exercise (PE), practised as daily recovery activities during lunch breaks, on perceived stress, general mental health, and immunoglobin A (IgA). A three-armed randomized controlled trial with 94 employees was conducted for five weeks including two follow-up sessions after one and six months. Daily practice lasted 30 min maximum. Perceived stress and general mental health questionnaires and saliva samples were used. There were significant differences in time factor comparing pre- and post-test of Perceived Stress Questionnaire (PSQ) both for PE [Mdiff = 0.10, SE = 0.03, *p* = 0.03], and for MM [Mdiff = 0.09, SE = 0.03, *p* = 0.03]. Moreover, there were significant differences of interaction factor when comparing MM vs. PE in total score at pre-post [F = −2.62 (6, 168.84), *p* = 0.02, ω^2^ = 0.09], favoring PE with medium and high effect sizes. Regarding General Health Questionnaire (GHQ) variable, practicing MM showed significant effects in time factor compared to pre-Fup2. No significant differences were found for IgA. Thus, practicing both MM and PE as recovery strategies during lunch breaks could reduce perceived stress after five weeks of practice, with better results for PE. Moreover, practicing MM could improve mental health with effects for 6 months.

## 1. Introduction

The lifestyle of advanced industrial societies is characterised by time pressure, competitiveness, job insecurity, 24/7 availability, sensory overload and multitasking in order to successfully fulfill a variety of roles [1]. Stress is a real health risk, which, in the short term, can trigger headaches, muscle tension, increased heart rate and blood pressure [2], emotional instability and irritability [3]. In the long term, stress may favour the onset of severe fatigue and exhaustion [4], anxiety and depression [5], cognitive problems and executive function issues [6], and chronic diseases [7].

More specifically, stress may entail the continuous adaptation of the neuroendocrine and immune systems [8]. In this respect, it has been observed that stress has a negative effect on salivary immunoglobulin A (sIgA) levels—among other immunological parameters [9,10]—the most abundant type of antibody found in the mucous membranes of the gastrointestinal and respiratory tracts [11,12,13,14]. This facilitates its laboratory analysis and thus the interpretation of IgA makes it an ideal parameter for conducting research on immune function [15,16,17]. Nevertheless, the relationship between sIgA responses and stressors is not linear, since sIgA increases as a response to both relaxation and stressors [18,19]. Therefore, it has been proposed that acute stress may potentially enhance the immune system as an adaptive response [20], while chronic stress produces a decline in immune functioning, associated with high susceptibility to infectious diseases [21,22].

Work and workplace-related issues are common sources of stress [23]. Work-related stress is associated with a decrease in productivity, increased absenteeism, accidents and injuries, mental illness, increased errors and poor performance, conflictive relations, somatic symptom disorders and even alcohol and drug abuse [24]. According to the International Labour Organization (2016) [25], stress is responsible for 30% of all work-related disorders. In the European Union, workplace stress has a negative impact on the wellbeing of 22% of the total labour force [26] and these data are expected to rise in the future [27,28].

All this implies significant financial costs to society. In the European Union, the cost of work-related depression is estimated at € 617 billion a year, including costs to employers resulting from absenteeism and presenteeism (€ 272 billion), loss of productivity (€ 242 billion), health care costs (€ 63 billion) and social welfare costs due to disability benefit payments (€ 39 billion) [29].

In this context, it seems crucial to find effective solutions. Sometimes, employees seek help to develop stress management strategies through psychotherapy, training courses, effective communication techniques, social support and relaxation exercises. At other times, companies implement changes to the work process, introducing flexible schedules, effective division of tasks, or a combination of both [30]. Richardson and Rothstein [31] conducted a meta-analysis to determine the effectiveness of stress management interventions in occupational settings, including 63 experimental studies, which concluded that relaxation and meditation techniques were the most popular individual interventions—with an average intervention length of 6.5 weeks and weekly 1 to 2-h sessions, and with an average effect size of 0.50 (Cohen’s d). Although cognitive-behavioural interventions seemed to be the most effective—with an average intervention length of 7.5 weeks and weekly 1 to 2-h sessions, and with an average effect size of 1.16—the popularity of relaxation and meditation techniques is due to the fact that they are more accessible and easy to implement than cognitive-behavioural interventions, where people need to become aware of their negative thoughts in order to modify their cognitive processes [32]. Along the same lines, a meta-analysis based on 43 primary studies and 22,822 workers [33] shows that cognitive-behavioural skills training, relaxation techniques and multimodal stress management training are positively related to psychological health, with a higher effect size in relaxation techniques.

Both practices can be considered recovery activities, within the framework of active rest strategies widely researched by Sonnentag and Fritz since 2005 [34]. According to the effort-recovery model, during the recovery process, an individual’s psycho-physiological system returns to its pre-stressor level by restoring energy and mental resources [35,36]. Thus, when recovery is insufficient, an individual has to invest additional effort at work, which may lead to accumulating tension in the long term [37,38].

Recovery processes can occur in various temporal and situational settings [39]. In this regard, research has analysed recovery during work shifts, free afternoons, weekends, holidays and sabbatical years, dividing these situations into two categories: internal recovery—during work time—and external recovery activities—outside work time [40]. However, few studies have focused on analysing recovery during lunch breaks, the period when employees stop working for approximately 30–60 min in the middle of their workday [41]. Some studies have observed that employees who report higher levels of psychological detachment, relaxation and control during this time of rest experience higher attention levels and less fatigue immediately after their break [42,43]. Additionally, employees who participated in restful activities during this period—socialising, taking a walk or a nap—felt more positive emotions and less negative emotions after their break [44,45]. On the other hand, employees who took part in work-related activities—e.g., preparing materials for a meeting—experienced a higher amount of negative emotions after their lunch break [46].

There is a critical need for effective methods to reduce work-related stress that may be employed by a large number of people, are easily accessible and have few side effects, and can be self-directed and practised with no time or location restrictions [47]. These factors are important since, on many occasions, stressed people are reluctant to consult a specialist or seek therapy due to the social stigma attached to mental ill health [48]. Given that “lack of time” is often reported as a reason for not adopting health-promoting behaviour [49], lunch breaks provide a unique opportunity to practice these type of activities.

Previous investigations show that mindfulness-based interventions (MBIs) and aerobic exercise could improve mental health and promote wellbeing [50,51,52], providing feasible and complementary alternatives to medical treatment [53,54,55].

Mindfulness meditation (MM) is a practice based on Buddhist traditions, which develops full attention and awareness through sitting meditation. It has rapidly gained popularity in the Western world due to its accessibility and easy practice. Jon Kabat-Zinn included this sitting meditation together with other practices—yoga and body scan—in his mindfulness-based stress reduction (MBSR) programme, an eight-week course, which trains full attention in everyday practice. Kabat-Zinn (1990) [56] defined mindfulness as “paying attention on purpose, in the present moment, and non-judgmentally, to the unfolding of experience moment to moment” (p. 23). Scientific evidence from analyses and meta-analyses has shown the results on the level of benefits of Mindfulness-based interventions (MBI), inter alia, over physical pain [57], depression [58], anxiety [59], stress and well-being [60], sleep [61], and inflammatory response [59]. In addition, approaches such as mindfulness training for introspection in the selfish–selfless spectrum are entering the mainstream of clinical care for managing pain, depression, and stress [62]. Regarding the employee’s mental health, significant results have been observed in decreased levels of emotional exhaustion (a dimension of burnout), and occupational stress, and also a significant increase in mindfulness, personal accomplishment, (occupational) self-compassion, quality of sleep, and relaxation [63].

On the other hand, physical exercise (PE) has been recognised for decades to maintain health, prevent illness and promote rehabilitation [64]. Its effectiveness in reducing stress and other related symptoms has been convincingly proven, and it is known to improve the state of mind and mitigate depression and anxiety, whether as part of a supervised or unsupervised programme [65,66,67]. However, exercise is still to be fully integrated in the treatment of mental ill health [68].

Research has been conducted to compare the implementation of MM and PE in the workplace, whether combined or individually, with interesting results [7,27,69]. Nevertheless, it has not been possible to find randomised controlled trials of MM and PE practised during lunch breaks by workers with medium stress levels. Therefore, in response to criticism by Farias and Wikholm [70] and Goldberg [71], the present study aims to compare the effects of MM with a bona fide practice like PE, and these effects, in turn, with those found in the inactive control group (CG). In this work a randomized controlled trial was carried out with pre-test, post-test and follow-up (one and six months after the post-test) in which in addition to the two experimental conditions indicated (MM and PE) an inactive control group (CG) was used. Therefore, the objective of the present study was to evaluate the effects of interaction, intragroup (temporal factor) and between-group in the following dependent variables: (a) perceived stress and subcomponents (social harassment-acceptance, overload, irritability-tension-fatigue, energetic joy, fear-anxiety and self-realization-satisfaction); (b) general mental health and subcomponents (coping strategies, self-esteem and stress); and (c) salivary immunoglobulin A.

We hypothesised that both interventions would reduce perceived stress and improve general mental health after five weeks of practice, but we did not have specific hypotheses about how long the effects would last or which intervention would be most preferable for each measure because of the lack of previous research comparing these interventions during lunch break. We also hypothesised that there would be no significant effects or differences on the IgA from the interventions in the medium term (5 weeks). sIgA was analysed as an objective variable to indicate whether and when the practice of MM and PE improves its response capacity. This parameter has been chosen in order to gain insight into a subject which seems to divide researchers. Some trials have found a clear relation between MM or PE and improved sIgA response when the latter is measured before and after the practice, with an interval of between 30 and 120 min [11,17]. Other studies have established longer intervals between measures, of weeks or even months depending on the pre-test and post-test of the intervention. In these cases, sIgA concentrations appear unaffected by the practice [15,16,72,73,74].

## 2. Materials and Methods

### 2.1. Participants

The analysis of the required sample size required 111 people, taking as reference the interaction factor with a power of 0.80, an alpha value of 0.05, and correcting the criterion of non-sphericity to 0.75 in order to reach a size of the effect between moderate and high (f = 0.39). Previous studies similar to ours (19) and WebPower package were used for R program (27).

The random sample consisted of 123 people. All subjects were, coincidentally, white (100%), and 67% were women with an average age of 46.81 (SD = 6.37). The participants of the MM group (*n* = 23, 76.7% women) had an average age of 47.4 ± 3.84. The participants of the PE group (*n* = 18, 60% women) had an average age of 47.77 ± 5.16. Finally, the participants of the control group (*n* = 22, 64.7% women) had an average age of 45.44 ± 8.66. All participants were employed by a multinational telecommunications company specialising in the service sector, had a permanent contract and a 9 a.m. to 6 p.m. work schedule, with a maximum lunch break of one and a half hours. Regarding their marital status, most of them were living with a partner: 61.1% (*n* = 44) were married and 20.8% (*n* = 15) had a stable non-marital partner. 93.1% (*n* = 67) of participants had completed university studies, including bachelor’s, master’s and doctoral degrees.

During lunch breaks, 51.4% had lunch with colleagues, 13.9% ate something quickly in order to catch up with work, 12.5% ate something quickly in order to deal with personal matters, 6.9% ate something quickly in order to have time to relax, take a walk, sunbathe or read a book, and 15.3% did not have a defined pattern.

### 2.2. Procedure

In order to obtain a homogeneous sample within the framework of the service sector, the selection of participants followed two inclusion criteria: mid-level professionals of the same organization, in this case team leaders, with medium levels of perceived stress at 0.35 (SD = 0.14) according to the Perceived Stress Questionnaire [75], with a total result of *n* = 123. That was chosen to ensure room for improvement and was derived from the probability scores found by Cohen and Janicki-Deverts [76]. The sample excluded workers (*n* = 29) who already practiced MM (*n* = 2) or PE (*n* = 20) more than once a week or who suffered some type of mental illness (*n* = 4 needed medication for depression and insomnia problems) or physical illness (*n* = 3 had chronic back pain).

The Human Resources department made the recruitment by sending emails with the invitation to participate in the trial to workers of the company with equal levels of responsibility. Once all participants were registered, and before agreeing to participate by signing the informed consent form, they were randomly allocated to the PE, the MM or control group conditions (ratio 1:1:1). Potential participants were given participant numbers upon enrolment with Stata software by independent research assistants who had no access to the randomization form. Participants received information on the condition to which they were allocated after the baseline measurements, composed of an online questionnaire for sociodemographic data, the Perceived Stress Questionnaire [75], the General Health Questionnaire [77] and a saliva sample collected at the end of the working day (at approximately 6 p.m.).

The study was approved by the Research Ethics Committee at Rey Juan Carlos University (No. 0709201711717, dated 28–09-2017) and registered at ClinicalTrial (NCT03728062). Figure 1 shows the CONSORT diagram of participation flow.

All participants completed the questionnaires and their saliva samples were collected at the end of the working day (at approximately 6 p.m.). Perceived stress and General mental health levels were measured at four different moments in time: pre-test, post-test and two follow-up measurements after one and six months. In the case of immunoglobulin A, saliva samples were collected three times (pre-test, post-test and follow-up measurement after a month). Additionally, each participant kept a record of his/her daily practice, which was supervised on a weekly basis.

Each of the three groups attended a four-hour information session. The first part of the session included a brief talk, which explained the effects of stress on mental and physical health, and the importance of recovery strategies. The second part focused on the general protocol of the intervention—length of the practice, collection of saliva samples, when to complete questionnaires, how to keep record of daily practice, etc. These two parts of the session were the same for the three groups. The third part of the session only involved the active intervention groups and was specific to each case. A certified and experienced MBSR instructor gave the participants of the MM GROUP training on mindfulness meditation practice. The participants of the PE group also received training, in this case given by a certified and experienced physical education instructor who explained the main aspects involved in aerobic exercise—outdoor or gym workouts, number of beats per minute, warm-up routine, etc.

The intervention took place during the five working days of five consecutive weeks, during which the two active groups practiced MM or PE during the lunch break (before having lunch), with equal time intervals of 15 min in the first week, 20 min in the second week, 25 min in the third week and 30 min in the fourth and fifth weeks. Each group had a weekly meeting with its instructor (MM and PE) who would introduce the weekly practice and clarify doubts. All participants kept a daily record of their practice in order to control that their adherence to the practice was at least 70%.

The MM group met with its certified MBSR instructor on Mondays, who explained the week’s meditation, based on Jon Kabat-Zinn’s MBSR Programme [56]. Participants were given instructions in writing and in audio format (mp3), so that they could practice meditation as a group—in a room set up by the company for this purpose—or individually in the place of their choice. The intervention followed a specific protocol: week 1, 15-min meditation based on breathing; week 2, 20-min meditation based on breathing and body awareness; week 3, 25-min meditation based on breathing, body awareness and hearing sensations; weeks 4 and 5, 30-min meditation based on breathing, body awareness and awareness of thoughts and emotions.

On the other hand, the PE group practiced aerobic exercise, which mainly consisted of running, training on an elliptical machine, rowing or cycling, outdoors or in the gym. Participants could choose the type of exercise they wanted to do and where to do it. However, the records show that most of them used the company’s gym. Participants started their exercise routine with a 5 to 7 min workout. They also had to maintain between 120 and 140 heartbeats per minute during their practice. The intervention was supervised by a certified instructor—bachelor’s degree in physical activity and sports sciences—and experienced physical activity trainer.

### 2.3. Instruments

#### 2.3.1. Perceived Stress Questionnaire (PSQ)

Stress levels were measured using the Perceived Stress Questionnaire designed by Levenstein et al. [75], where the PSQ index is obtained using the formula PSQ = (raw score−30)/90, following the Spanish version validated by Sanz-Carrillo, García-Campayo, Rubio, Santed, and Montoro [78], and with results varying from 0 to 1. This instrument is composed of 30 items—e.g., “You feel tired”, “You feel that too many demands are being made on you” or “You find yourself in situations of conflict”—with Likert-type responses ranging from 1 (rarely) to 4 (almost always), and with the following six subscales: harassment-social acceptance (sample Cronbach α = 0.72), overload (sample Cronbach α = 0.69), irritability-tension-fatigue (sample Cronbach α = 0.80), energy-joy (sample Cronbach α = 0.72), fear-anxiety (sample Cronbach α = 0.18), and self-realisation-satisfaction (sample Cronbach α = 0.49); total PSQ (sample Cronbach α = 0.89). The internal consistency of the test for the Spanish population is α = 0.87.

#### 2.3.2. General Health Questionnaire (GHQ-12)

General mental health was measured using the short version of the General Health Questionnaire [77] validated for Spanish population by Rocha, Pérez, Rodríguez-Sanz, Borrell, and Obiols [79]. This questionnaire is a mental health screening test composed of 12 items—6 positive and 6 negative—such as “Have you recently felt capable of making decisions about things?”, “Have you felt that you could not overcome your difficulties?” or “Have you been able to enjoy your normal day-to-day activities?”. The Likert-type responses range from 0 (a lot less than usual) to 3 (more than usual). General health perception is evaluated through three subscales of Coping Strategies (sample Cronbach α = 0.82), Self-esteem (sample Cronbach α = 0.86), Stress (sample Cronbach α = 0.42) and total GHQ (0.82) [77]. GHQ-12′s reliability varies from 0.82 to 0.86 [75].

#### 2.3.3. Salivary Immunoglobulin A (sIgA)

Salivary IgA response levels were determined through the collection of saliva samples in Salivette^®^ tubes (Sarstedt, Rommersdolf, Germany). The saliva samples were frozen and kept at −20°C immediately after arrival at the university laboratory, until they were sent to an external laboratory for analysis. SIgA levels were measured by nephelometry (BN-II), using the reagent OSAR15 anti-IgA (Dade Behring), with a sensitivity of 0.2 mg/dL. SIgA levels are expressed in mg / dL.

### 2.4. Data Analyses

Overall, three sets of statistical analysis were conducted. First, data were explored to verify normal distribution. Secondly, a descriptive analysis of the sample was conducted comparing, in addition, the experimental groups in the sociodemographic and dependent variables on the baseline, running the Chi-square test (qualitative variables), together with a univariate ANOVA for the analysis of quantitative variables. Thirdly, the analysis of the effects of the interventions is carried out through an intention-to-treat analysis by adjusting a mixed linear model (MLM) using the maximum restricted likelihood method for the group, time and interaction factors (group x time). These models have several advantages over the general linear model, allowing the assumptions of independence to be relaxed, especially important in designs with repeated measures, as well as that of equality of variance, since it allows working with different covariance structures [80].

On the other hand, it allows all available data to be used, which avoids having to use imputation methods for missing data. Likewise, effect sizes (ES) are reported by ω 2 in the analyses of the main effects, and d for the analysis of multiple comparisons [81]. For each significant main effect, multiple post-hoc comparisons are made by adjusting the error rate to the inflation produced by multiple variables using the Holm—Bonferroni’s procedure [82].

## 3. Results

Table 1 shows the set of analysed sociodemographic variables and comparisons on the baseline, where the absence of significant differences between the groups can be verified. The Appendix A shows the results of the analysis by protocol (Appendix A), whose results are similar to the intention-to-treat analysis (Table 1, Table 2 and Table 3).

### 3.1. Main Effects

First, we analysed the *main effects* of the intragroup (Time), intergroup (Group) and Time X Group interaction factors (see Table 2 and Appendix A).

Regarding the *interaction factor*, since the interventions are compared at different time points with respect to the pre-test, statistically significant differences have been obtained with moderate effect sizes in the PSQ variable, in the total scores [f = 2.62 (6, 168.84), *p* = 0.02, ω ^2^ = 0.09], and in harassment-social acceptance dimensions [F = 3.63 (6, 160.22), *p* < 0.01, ω ^2^ = 0.14], overload [F = 2.34 (6, 169), *p* < 0.01, ω ^2^ = 0.08], and irritability-tension-fatigue [f = 2.25 (6, 171.61), *p* = 0.04, ω ^2^ = 0.07]. However, no significant differences were found for the interaction factor in GHQ or IgA.

As for the *time factor*, since the intragroup efficacy of the different interventions is evaluated, apart from the energy-joy dimension (PSQ) and the sIgA variable, the set of analysed variables showed significant differences (all < 0.01), with effect sizes ranging from moderate to high in PSQ (total score and subscales), and in GHQ (total score and subscales).

The *group factor* analysis compares the different interventions at each time point, regardless of the baseline. It has not shown statistically significant differences in any variable.

### 3.2. Simple Effects

The *main effects* seen above allow us to determine on which factor the significant effects (Time, Group or Interaction) rest, but it does not allow us to discriminate between on which elements of the possible comparisons the differences occur within each factor. This issue is addressed by analysing the *simple effects* (multiple comparisons) with the corresponding correction of the type I error. These comparisons are made on those factors that have significant results in the variables previously analysed in the main effects (Appendix A).

Table 3 shows the simple effects of the *interaction factor* for PSQ, derived from comparisons between each pair of elements of the group factor (MM vs. PE; MM vs. CG; PE vs. CG) at each time point with respect to the baseline. The results show significant differences in most pre-post comparisons with medium and high effect sizes. Specifically, the total PSQ score for MM vs. PE was (t = −2.26, *p* = 0.02, d = 0.59) and for PE vs. CG was (t = 3.72, *p* < 0.01, d = 0.94). In the case of the subscales, it gave the following significant results: harassment-social acceptance for MM vs. PE (t = −2.59, *p* = 0.01, d = 0.68) and for PE vs. CG (t = 4.15, *p* < 0.01, d = 1.05); overload for MM vs. CG (t = 2.36, *p* = 0.02, d = 0.60) and for PE vs. CG (t = 3.07, *p* < 0.01, d = 0.78); and irritability-tension-fatigue for MM vs. PE (t = −2.06, *p* = 0.04, d = 0.54) and for PE vs. CG (t = 3.20, *p* < 0.01, d = 0.81).

Finally, we have evaluated the intragroup efficacy of MM and PE interventions derived from the significant major effects of the *time factor*. As seen in Table 4, the set of analysed variables showed significant differences (all <0.01), with effect sizes ranging from moderate to high for MM in GHQ (total score) and coping dimension, and in the pre-Fup1 comparisons for self-esteem and stress dimensions.

In addition, correlations between the amount of final practice and the degree of recovery in PSQ, GHQ and IgA (pre-test—post-test change scores) were performed without significant differences (Appendix A).

## 4. Discussion

The present study proposes that both MM and PE practised during lunch breaks could be considered recovery activities from stress during lunch time since they significantly improve perceived stress of employees, where medium and high effect sizes are observed after 5 weeks of practice (Table 3). However, its effects do not last long, since only one aspect of the PSQ, irritability, lasts after six months of the practice. Comparing both activities, physical exercise is proposed as the strategy with better results obtained in perceived stress (Figure 2). PE seems to have larger effect sizes than MM, especially on the PSQ dimensions of irritability-tension-fatigue, overload and harassment-social acceptance. This means that workers who practice PE feel there is less conflict in their everyday activities, less frustration, more security and protection, less work pressure, and that they can look to the future and cope with responsibilities better. In this regard, this study coincides with other research such as Van der Zwan et al., [69], which also established larger effect sizes for PE than MM regarding the reduction of stress. Last, we cannot find an explanation for the slight improvement in the results of the control group on perceived stress. However, these are not significant results.

Regarding mental health (Table 4), we have observed significant differences in the intragroup effects of the mindfulness group, and they last at least up to six months after the end of the 5-week practice. However, in general terms, no great differences have been observed between the results of MM and PE regarding mental health. This may be due to the fact that mindfulness meditation and aerobic exercise share similar cognitive mechanisms, such as the capacity to concentrate on only one task and sensory awareness of the body, which may improve overall mental health, including the adaptive responses to stress [83,84,85]. Although no significant differences have been found when comparing MM and PE, some interesting differences have been observed within some of the specific dimensions of the questionnaires. Thus, MM seems to have an impact on the overall improvement of mental health and especially on coping capacity, and consequently on concentration, decision making, the ability to enjoy everyday activities, deal with problems adequately and, in general, “feel reasonably happy” [77]. The study is, therefore, consistent with scientific analyses, which indicated that MM and PE could be considered non-pharmacological treatments for the improvement of mental health, including adaptive responses to stress [69,86]. More specifically, our article is in line with others, which highlight the advantages of MM over PE regarding mental health [83,84,87,88].

Finally, regarding IgA, neither one of these practices involve an improvement in sIgA response capacity, neither intragroup nor intergroup. As in the Song & Baicker study [89], we have found positive results in self-reported health measures, but there were not significant differences in measures with biomarkers such as IgA. With regard to sIgA, as mentioned previously, no significant differences have been observed between the practice of MM and PE regarding the immunocompetence of IgA, in any of the conducted analyses. Thus, our results are in line with those of other authors who also compared the effects of IgA in MM and PE after inoculation with the flu vaccine [16,72,87,90]. They are also consistent with trials that only analysed IgA in MM, such as the systematic review undertaken by Black and Slaich [15]. On the other hand, regarding PE, research undertaken to establish whether sIgA response capacity varied with intense exercise [74] or in athletes [91] also yielded no significant results.

However, our results contradict those of other studies which did find significant results, such as the study conducted by Bellosta-Batalla et al., [11] on MM and self-compassion, or research focusing on individuals who practiced MM and were inoculated with the flu vaccine [17], or on cancer patients [92]. It is worth noting that in these studies, sIgA was measured just before and after each intervention session, with no more than 2 h in between each saliva sample collection. However, in studies that did find significant results, the lapse of time between sample collections was of weeks—as in the case of our study—or even months. So, it may be concluded that, according to this investigation, practising PE or MM yields no benefits in the medium term.

We are not aware of any studies on mindfulness or physical exercise practised as recovery activities during lunch breaks, although there is evidence of comparisons of other practices, such as progressive muscle relaxation and small-talk break groups [42], exposure to nature and relaxation, and relaxation and park walks [93]. Therefore, this study offers an innovative perspective to the scientific literature on internal recovery at work, supplementing the results of other studies on the practice of mindfulness meditation during work hours [94,95]. In this regard, the most important finding is that, after the third week of practicing PE, there is a significant improvement in energy levels and less fatigue at the end of the working day, which is characteristic of recovery activities. Thus, workers feel more rested, less irritable, calmer, happier, less hurried, and with more mental energy and time to enjoy themselves [75]. However, it is important to remember that these effects disappear when PE stops being practiced.

We agree with Edwards’ systematic review [94] on the compared effects of MM and PE in that it is important to know the specific objectives and benefits of each of the two interventions in order to optimise time. Nevertheless, we believe motivation is essential and, therefore, each individual should spend time on the practice of their choice [95].

The strengths of this study include a three-armed randomised controlled design—with up to 8 repeated measurements—the use of scales validated for the Spanish population, statistical methods and biological measurements, with medium to large effect sizes in the results of subjective variables. However, the study involves certain limitations. For example, the sample cannot be considered representative of the population as a whole, since it consists of university-educated Caucasian workers employed in a very specific sector. Some of the study’s data were obtained from self-reported measures, subject to social desirability bias. Taking into account the limitations and results of this study, it would be relevant for future investigations to evaluate full attention and exercise programmes with a sample that includes higher racial/ethnic diversity, and different socioeconomic backgrounds and professional sectors. It would also be appropriate to increase the timespan of the practice to, at least, three months since most studies only entail eight weeks, and perhaps this is not enough time for significant effects of mindfulness intervention; since this is an eminently cognitive practice, it may require more time than physical exercise to learn, assimilate and have significant effects in the short and medium term.

Finally, we have been informed that two small groups with 74% of participants of the MM group and 45% of the PE group, spontaneously developed from the original intervention groups, continued practising MM and PE after the end of the study. We consider that it would be interesting to conduct future studies on the effects of these “spontaneous” groups the interpersonal relationships in the workplace, in order to analyse whether these practices are beneficial in promoting better working environments. Previous research has shown the associated effects of social commitment on mental health [94,96].

The practical implications of this study derive from the results discussed. Additionally, the possibility of conducting short effective internal recovery practices during the lunch break may make these activities suitable for busy employees who have a higher workload, an important limitation in this type of intervention [46]. Another relevant practical implication is the high level of autonomy involved in both MM and PE, especially after three weeks, as established by the development of informal groups, which continued with the practice after the intervention had ended. Furthermore, the autonomy perceived during the lunch break maximises the recovery effect of the activities [43]. Thus, the energising effect of MM and PE during lunch break can be supplemented with minibreaks during work hours in order to promote employees’ positive sense of job meaning and work experiences [46]. In short, the amount of resources invested by organisations in facilitating recovery activities is low compared with the results these practices yield in terms of employee wellbeing, health, performance and satisfaction.

## 5. Conclusions

The present study provides preliminary evidence on Mindfulness Meditation and Physical Exercise as recovery strategies during lunch breaks in the workplace to reduce perceived stress after five weeks of practice, with better results for Physical Exercise compared to Mindfulness Meditation. However, practicing Mindfulness Meditation at lunch break could improve global mental health in the workplace with effects lasting for six months. No significant results were found for IgA in saliva neither in intragroup nor between-groups comparisons.

## Figures and Tables

**Figure 1 ijerph-17-02839-f001:**
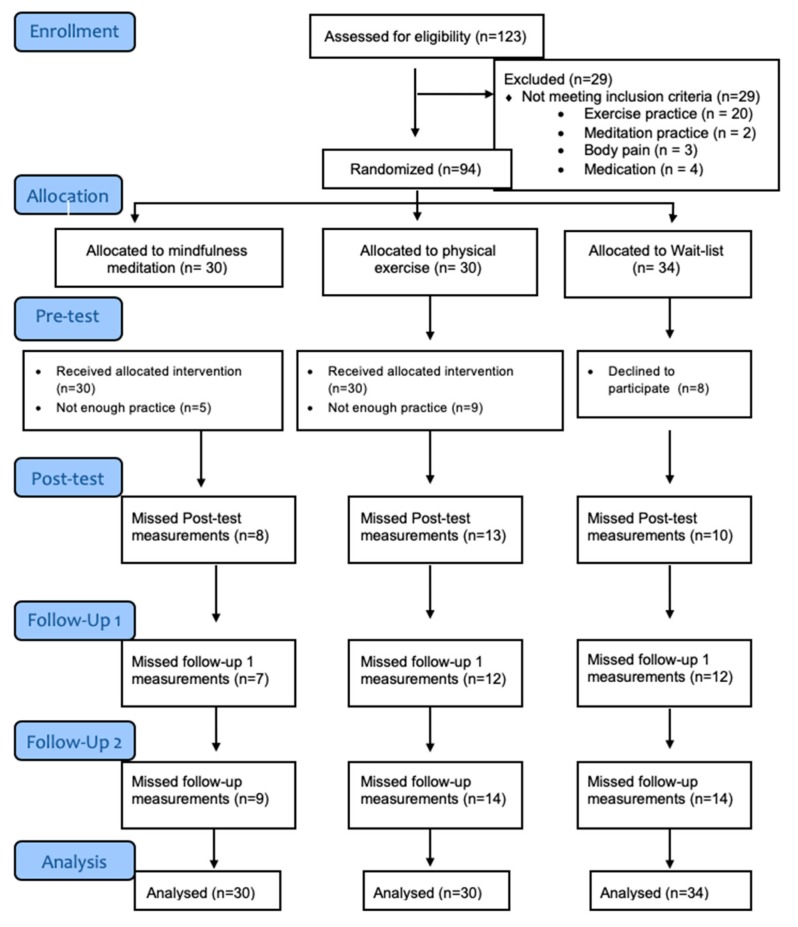
Consolidated Standards of Reporting Trial (CONSORT) diagram for a Randomized Controlled Trial of mindfulness meditation, physical exercise and control group.

**Figure 2 ijerph-17-02839-f002:**
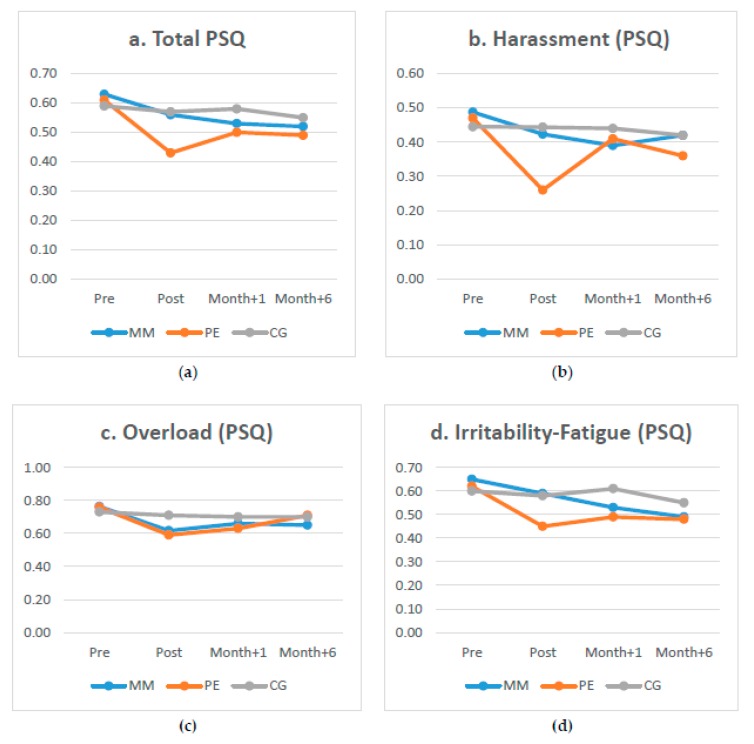
(**a**) Perceived Stress Questionnaire (PSQ) (**b**) Harassment-Social acceptance of the Perceived Stress Questionnaire (PSQ) (**c**) Overload dimension of the Perceived Stress Questionnaire (PSQ) (**d**) Irritability-Tension-Fatigue dimension of the Perceived Stress Questionnaire (PSQ).

**Table 1 ijerph-17-02839-t001:** Descriptive analysis and comparisons of sociodemographic and dependent variables on the baseline.

Variables	MM (*n* = 30)	PE (*n* = 30)	CG (*n* = 34)	Total (*N* = 94)	
	**M (SD)**	**M (SD)**	**M (SD)**	**M (SD)**	**F/*p*^1^**
Age	47.40 (3.84)	47.77 (5.16)	45.44 (8.66)	46.81 (6.37)	F = 1.26, *p* = 0.29
PSQ ^2^					
Harassment-SA	0.48 (0.20)	0.47 (0.15)	0.45 (0.15)	0.47 (0.17)	F = 0.23, *p* = 0.80
Overload	0.76 (0.19)	0.76 (0.13)	0.73 (0.15)	0.75 (0.16)	F = 0.63, *p* = 0.54
Irritab.-Fatigue	0.65 (0.21)	0.62 (0.14)	0.60 (0.14)	0.62 (0.16)	F = 0.71, *p* = 0.49
Energy-joy	0.55 (0.19)	0.55 (0.19)	0.56 (0.15)	0.56 (0.17)	F = 0.03, *p* = 0.97
Fear-anxiety	0.53 (0.26)	0.47 (0.18)	0.47 (0.23)	0.49 (.22)	F = 0.71, *p* = 0.49
Self-realisation	0.57 (0.23)	0.56 (0.15)	0.53 (0.20)	0.55 (0.19)	F = 0.44, *p* = 0.64
Total	0.63 (0.18)	0.61 (0.11)	0.59 (0.12)	0.61 (0.14)	F = 0.51, *p* = 0.60
GHQ ^3^					
Coping	6.96 (3.31)	6.50 (1.70)	7.40 (2.86)	6.95 (2.69)	F = 0.84, *p* = 0.44
Self-esteem	5.07 (3.91)	3.63 (2.58)	3.67 (2.93)	4.10 (3.20)	F = 1.92, *p* = 0.15
Stress	5.11 (2.04)	4.77 (1.36)	4.43 (1.77)	4.76 (1.74)	F = 1.09, *p* = 0.34
Total	17.14 (7.20)	14.90 (4.71)	15.50 (6.32)	15.82 (6.14)	F = 1.03, *p* = 0.36
IgA	2.66 (1.41)	3.05 (1.09)	2.61 (1.07)	2.76 (1.21)	F = 0.84, *p* = 0.44
	***n* (%)**	***n* (%)**	***n* (%)**	***N* (%)**	**X^2^/*p*^1^**
Sex					X^2^ = 2.01, *p*= 0.36
Woman	23 (76.7%)	18 (60.0%)	22 (64.7%)	63 (67.0%)	
Man	7 (23.3%)	12 (40.0%)	12 (35.3%)	31 (33.0%)	
Marital status					X^2^ = 10.56, *p* = 0.23
Married	17 (56.7%	25 (83.3%)	17 (50.0%)	59 (62.8%)	
Stable partner	7 (23.3%)	2 (6.7%)	9 (26.5%)	18 (19.1%)	
Single	3 (10.0%)	1 (3.3%)	2 (5.9%)		
Separated/Divorced	3 (10.0%	2 (6.7%)	5 (14.7%)	10 (10.6%	
Widower	0 (10.0%)	0 (10.0%)	1 (2.9%)	1 (1.1%)	
Level of education					X^2^ = 6.40, *p* = 0.38
Second. education	1 (3.3%)	2 (6.7%)	2 (5.9%)	5 (5.3%)	
Bachelor’s degree	19 (63.3%)	12 (40.0%)	13 (38.2%)	44 (46.8%)	
Master’s degree	10 (33.3%)	53.3 (17.0%)	18 (52.9%)	44 (46.8%)	
Doctoral degree	0 (0.0%)	0 (0.0%)	1 (2.9%)	1 (1.1%)	

^1^*p* < 0.05. ^2^ Perceived Stress Questionnaire. ^3^ General Health Questionnaire.

**Table 2 ijerph-17-02839-t002:** Results of the mixed linear model (MLM) for the main effects—group, time and interaction factors.

Variable	Interaction			Time			Group			α
	F	*p*	ω ^2^	F	*p*	ω ^2^	F	*p*	ω ^2^	
GHQ										
Coping	1.37	0.23	0.02	11.55	<0.01 *	0.25 ^##^	2.94	0.06	0.04	0.82
Self-esteem	1.21	0.30	0.01	9.25	<0.01 *	0.21 ^##^	1.28	0.28	0.01	0.86
Stress	1.49	0.18	0.03	9.16	<0.01 *	0.21 ^##^	0.70	0.50	<0.01	0.42
Total	1.74	0.11	0.04	16.67	<0.01 *	0.33 ^##^	0.54	0.58	<0.01	0.82
**PSQ**										
Harassment-SA	3.63	<0.01 *	0.14 ^#^	6.96	<0.01 *	0.22 ^##^	1.79	0.17	0.02	0.72
Overload	2.34	<0.01 *	0.08 ^#^	8.54	<0.01 *	0.19 ^##^	0.88	0.42	<0.01	0.69
Irritab.-fatigue	2.25	0.04 *	0.07 ^#^	8.66	<0.01 *	0.20 ^##^	1.92	0.15	0.02	0.80
Energy-joy	0.69	0.66	<0.01	1.59	0.19	0.02	1.24	0.29	0.01	0.59
Fear-anxiety	1.42	0.21	0.03	6.85	<0.01 *	0.16 ^##^	0.28	0.76	<0.01	0.18
Self-realisation	1.25	0.28	0.01	5.17	<0.01 *	0.12 ^#^	1.57	0.21	0.01	0.47
Total	2.62	0.02 *	0.09 ^#^	9.13	<0.01 *	0.21 ^##^	2.06	0.13	0.02	0.89
**IgA**	0.77	0.55	<0.01	0.88	0.42	<0.01	0.98	0.38	<0.01	

* *p* < 0.05. ^#^ ω ^2^. Moderate effect size. ^##^ ω ^2^. High effect size.

**Table 3 ijerph-17-02839-t003:** Analysis of simple effects—interaction factor (PSQ).

Variable		Groups													
		MM ^1^ vs. PE ^2^						MM vs. CG ^3^					PE vs. CG		
		t	p	d	I.C.95%		t	p	d	I.C. 95%		t	p	d	I.C. 95%
**Haras.**															
	**Pre-Post**	−2.59	0.01 *	0.68	[−0.23, −0.03]	1.55	0.12	0.40	[−0.02, 0.16]		4.15	<0.01 *	1.05	[0.10, 0.30]	
	**Pre-Fup 1**	0.43	0.67	0.11	[−0.10, 0.16]		1.62	0.11	0.41	[−0.02, 0.23]		1.10	0.27	0.28	[−0.06, 0.21]
	**Pre-Fup 2**	−0.35	0.72	0.09	[−0.16, 0.11]		0.82	0.41	0.21	[−0.08, 0.19]		1.15	0.25	0.29	[−0.06, 0.21]
**Overl.**															
	**Pre-Post**	−0.77	0.44	0.20	[−0.14, 0.06]		2.36	0.02 *	0.60	[0.02, 0.21]		3.07	<0.01 *	0.78	[0.05, 0.25]
	**Pre-Fup 1**	−0.87	0.38	0.23	[−0.19, 0.07]		1.32	0.19	0.33	[−0.04, 0.21]		2.07	0.04 *	0.52	[0.01, 0.28]
	**Pre-Fup 2**	0.88	0.38	0.23	[−0.07, 0.19]		1.34	0.18	0.34	[−0.04, 0.21]		0.42	0.67	0.11	[-0.10, 0.16]
**Irritab.**															
	**Pre-Post**	−2.06	0.04	0.54	[−0.20, −0.01]		1.12	0.26	0.28	[−0.04, 0.15]		3.20	<0.01 *	0.81	[0.06, 0.26]
	**Pre-Fup 1**	−0.21	0.83	0.05	[−0.14, 0.12]		1.58	0.11	0.40	[−0.02, 0.23]		1.68	0.09	0.43	[−0.02, 0.25]
	**Pre-Fup 2**	0.50	0.61	0.13	[−0.10, 0.17]		1.87	0.06	0.47	[−0.01, 0.26]		1.32	0.19	0.33	[−0.04, 0.22]
**Total**															
	**Pre-Post**	−2.26	0.02	0.59	[−0.19, −0.01]		1.44	0.15	0.36	[−0.02, 0.14]		3.72	<0.01 *	0.94	[0.07, 0.25]
	**Pre-Fup 1**	−0.38	0.71	0.10	[−0.13, 0.09]		1.39	0.17	0.35	[−0.03, 0.19]		1.66	0.10	0.42	[−0.02, 0.22]
	**Pre-Fup 2**	0.31	0.76	0.08	[−0.10, 0.14]		1.47	0.14	0.37	[−0.03, 0.20]		1.13	0.26	0.29	[−0.05, 0.18]

* *p* < 0.05. ^1^ Mindfulness Meditation Group. ^2^ Physical Exercise Group. ^3^ Control Group.

**Table 4 ijerph-17-02839-t004:** Comparisons of simple effects—time factor GHQ.

Variable	Group	Comparisons	M Diff ^1^	S.E.	*p*	C.I. 95%	
						LLCI ^2^	ULCI ^3^
**Total**	Mindfulness	Pre-Post-test	4.64	1.24	0.00 *	1.28	8.00
		Pre-Fup 1	5.48	1.20	0.00 *	2.23	8.73
		Pre-Fup 2	4.74	1.48	0.01 *	0.71	8.77
	Physical exercise	Pre-Post-test	3.05	1.35	0.16	−0.62	6.71
		Pre-Fup 1	3.14	1.31	0.11	−0.40	6.69
		Pre-Fup 2	0.10	1.62	0.99	−4.30	4.50
**Coping**	Mindfulness	Pre-Post-test	3.24	0.60	0.00 *	1.61	4.87
		Pre-Fup 1	3.24	0.66	0.00 *	1.46	5.02
		Pre-Fup 2	3.24	0.79	0.00 *	1.09	5.39
	Physical exercise	Pre-Post-test	1.29	0.65	0.32	−0.49	3.06
		Pre-Fup 1	1.43	0.72	0.30	−0.52	3.37
		Pre-Fup 2	0.33	0.86	0.99	−2.02	2.68
		Pre-Fup 2	-0.24	0.70	0.99	−2.14	1.67
**Self-esteem**	Mindfulness	Pre-Post-test	0.88	0.63	0.99	−0.84	2.60
		Pre-Fup 1	1.60	0.49	0.01 *	0.28	2.92
		Pre-Fup 2	0.78	0.64	0.99	−0.97	2.53
	Physical exercise	Pre-Post-test	1.29	0.69	0.40	−0.59	3.16
		Pre-Fup 1	1.24	0.53	0.13	−0.20	2.68
		Pre-Fup 2	−0.24	0.70	0.99	−2.14	1.67
**Stress**	Mindfulness	Pre-Post-test	0.80	0.44	0.43	−0.39	1.99
		Pre-Fup 1	1.20	0.38	0.01 *	0.17	2.23
		Pre-Fup 2	1.04	0.46	0.16	−0.21	2.29
	Physical exercise	Pre-Post-test	1.19	0.48	0.09	−0.10	2.48
		Pre-Fup 1	0.95	0.41	0.14	−0.17	2.07
		Pre-Fup 2	0.00	0.50	0.99	−1.36	1.36

* *p* < 0.05. ^1^ difference between means. ^2^ lower limit class interval. ^3^ upper limit class interval.

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
