# Peer review of "Mindfulness versus Physical Exercise: Effects of Two Recovery Strategies on Mental Health, Stress and Immunoglobulin A during Lunch Breaks. A Randomized Controlled Trial"

_ijerph, 2020, doi:10.3390/ijerph17082839_

Round 1
Reviewer 1 Report
Dear authors,
Thank you for submitting this study. Overall, I think your study add value to the process to build evidence on effective methods to deal with stress. I have just some minor questions and suggestions to further improve your paper.
Do you have any information on baseline mental health status and baseline perceived stress level? Does the Table 1 illustrate the baseline data, or did I misunderstand that? Maybe, this could be further declared in the heading for the Table.
The figure a,b,c and d shows post intervention values. Do you know if some of the participants did continue to use MM or PE during this period, or if they all stopped?
Author Response
RESPONSES TO REVIEWER 1
Point 1. Do you have any information on baseline mental health status and baseline perceived stress level? Does the Table 1 illustrate the baseline data, or did I misunderstand that? Maybe, this could be further declared in the heading for the Table.
Response 1. The information on the baseline mental health status of the sample is indeed found in Table 1, entitled ‘Table 1. Descriptive analysis and comparisons of sociodemographic and dependent variables on the baseline.’ In it, you will see indicated the total punctuation and that of each of the subscales, in the form of average and standard deviation, for each of the three groups. For example, it shows how, in the total GHQ score, the MM Group has 17.14 (7.20), the PE group 14.90 (4.71), the CG Group 15.50 (6.32) and the total sample 15.82 (6.14). Similarly, the baseline scores of the PSQ and its subscales appear, also in the form of mean and standard deviation. Thus, in the MM group .63 (.18), in the PE group .61 (.11), in the CG group .59 (.12), and in the total sample .61 (.14). All these scores are also reflected in Table 2 of the supplementary material.
Point 2. The figure a,b,c and d shows post intervention values. Do you know if some of the participants did continue to use MM or PE during this period, or if they all stopped?
Response 2. Thanks to the record of the daily practice that we requested during the six months after the intervention, we were able to see that, once the intervention ended, the practice was negligible. During the first month it was 8%, and during the following months about 4%. Therefore, we do not believe that it could have altered the scores after the intervention.
Reviewer 2 Report
The literature review is extensive and well referenced. You could benefit by including references to articles identifying the sociological and interpersonal, cooperativity improvements that PE and MM produce, for instance:
- https://dx.doi.org/10.3389%2Ffpsyg.2018.00575
- https://dx.doi.org/10.1371%2Fjournal.pone.0191332
It is unclear if the group assignments were truly randomized. The Methods seems to indicate that participants were assigned sequentially based on the time at which they joined the study, which could have produced significant sorting based on factors such as motivation and enthusiasm. Ideally subjects would have been assigned identifiers and a database used to randomly select 1/3 of the subjects for each group. This should be clarified in the Methods and discussed as a potential limitation.
The discussion of limitations includes a description of the participants as not matching the general population in a number of ways, including race. However, race was not indicated within the manuscript demographic Table. In order for the reader to evaluate this limitation, these factors must be presented, or described as being observational in nature and not specifically collected.
Table 1 and 2 need significant clarification. For instance, Table 1 in the “Widower’s” row lists an extra number and percentage in a far right column defined as the statistical results column. This number needs to be correctly ascribed. Table 2 appears to be the output of a mixed model ANOVA, and includes Time and Group which appear to be described as being part of Supplementary Tables 3 and 4 according to the text below the Table. It is unclear how this data was analyzed, as the results appear to be comparing only two groups. Were these two groups the interventional groups (MM and PE) pre- v. post-intervention timepoints? Were the intervention groups combined and the table is comparing Interventions v. Waitlist group? Please clarify. In Table 3 a value is missing in the MM v. CG comparison of irritability. Please clarify.
There are two “Figure 1” figures in your article. The second Figure 1 uses a different acronym legend, “MF” presumably for “MM.” Please clarify. Please include standard deviation error bars on these graphs in order for the reader to be able to interpret them, and consider adding notations for significantly different comparisons.
Your Discussion asserts that MM improves mental health better than PE, but I disagree and see no data that consistently supports that interpretation. Please clarify, and reference the supporting data.
The limitations of your study are not sufficiently addressed. How was adherence to the intervention regimen controlled or monitored? The waitlist control group appeared to trend toward improvement in some measures; did the control group participate in weekly lunchtime interactions of a controlled nature? Was there evidence that they were taking part in an intervention? Please discuss what a proper control group would look like and how closely your group matches that ideal. You mention that your intervention groups continued their interventions after the study treatment period. Is it possible the control group members created their own intervention while “waiting?”
Based on the data presented, I cannot agree that your conclusion that MM improving global mental health is supported by your results. Please clarify in your discussion what data supports this conclusion.
Author Response
RESPONSES TO REVIEWER 2
Point 1. The literature review is extensive and well referenced. You could benefit by including references to articles identifying the sociological and interpersonal, cooperativity improvements that PE and MM produce. Response 1: Thank you for bringing us these two new and interesting references. We have included both of them on lines 111-116 of the text. In addition, we have updated the new list of references, including 62 and 63 respectively.
Point 2. It is unclear if the group assignments were truly randomized. The Methods seems to indicate that participants were assigned sequentially based on the time at which they joined the study, which could have produced significant sorting based on factors such as motivation and enthusiasm. Ideally subjects would have been assigned identifiers and a database used to randomly select 1/3 of the subjects for each group. This should be clarified in the Methods and discussed as a potential limitation.
Response 2: (lines 179-180) Thank you for your assessment. The expression "immediately upon registration" is incorrect. It should be replaced by “once all participants were registered” to avoid creating confusion for the reader. In the process, the 94 subjects were randomized by an assigned code (e.g. MM101, PE101 and CG101), and then the Stata software randomly assigned each of them to a group one by one. In no way was it assigned sequentially, but once they were all duly registered.
Point 3. The discussion of limitations includes a description of the participants as not matching the general population in a number of ways, including race. However, race was not indicated within the manuscript demographic Table. In order for the reader to evaluate this limitation, these factors must be presented, or described as being observational in nature and not specifically collected.
Response 3: We understand that it would be sufficient to add in the description of the 'Participants' that 'all subjects were, coincidentally, white.' We did not intend for this information to be reflected on the demographic table. Given that 100% of the participants fell within the same racial group, we considered that we could eliminate this data, so as not to make the table overly extensive. Thus, line 155 includes ‘The random sample consisted of 123 people. All subjects were, coincidentally, white (100%)…’.
Point 4.a. Table 1 and 2 need significant clarification. For instance, Table 1 in the “Widower’s” row lists an extra number and percentage in a far right column defined as the statistical results column. This number needs to be correctly ascribed.
Response 4.a.: The error on Table 1 has been corrected. Thank you for your observation.
Point 4.b. Table 2 appears to be the output of a mixed model ANOVA, and includes Time and Group which appear to be described as being part of Supplementary Tables 3 and 4 according to the text below the Table. It is unclear how this data was analyzed, as the results appear to be comparing only two groups. Were these two groups the interventional groups (MM and PE) pre- v. post-intervention timepoints? Were the intervention groups combined and the table is comparing Interventions v. Waitlist group? Please clarify.
Response 4b: We have changed the wording for the ‘Main effects’ (section 3.1; line 287) and the ‘Simple effects’ (section 3.2). It would look like this:
3.1. Main effects
First, we analysed the main effects of the intragroup (Time), intergroup (Group) and Time X Group interaction factors (see Table 2).
Regarding the interaction factor, since the interventions are compared at different time points with respect to the pre-test, statistically significant differences have been obtained with moderate effect sizes in the PSQ variable, in the total scores [f = 2.62 (6, 168.84), p = .02, ?2 = .09], and in harassment-social acceptance dimensions [F = 3.63 (6, 160.22), p <.01, ?2 = .14], overload [F = 2.34 (6, 169), p <.01, ?2 = .08], and irritability-tension-fatigue [f = 2.25 (6, 171.61), p = .04, ?2 = .07]. However, no significant differences were found for the interaction factor in GHQ or IgA.
As for the time factor, since the intragroup efficacy of the different interventions is evaluated, apart from the energy-joy dimension (PSQ) and the sIgA variable, the set of analysed variables showed significant differences (all <.01), with effect sizes ranging from moderate to high in PSQ (total score and subscales), and in GHQ (total score and subscales).
The group factor analysis compares the different interventions at each time point, regardless of the baseline. It has not shown statistically significant differences in any variable.
3.2 Simple effects
The main effects seen above allow us to determine on which factor the significant effects (Time, Group or Interaction) rest, but it does not allow us to discriminate between which elements of the possible comparisons the differences occur within each factor. This issue is addressed by analysing the simple effects (multiple comparisons) with the corresponding correction of the type I error rate. These comparisons are made on those factors that have significant results in the variables previously analysed in the main effects.
Table 3 shows the simple effects of the interaction factor for PSQ, derived from comparisons between each pair of elements of the group factor (MM vs. PE; MM vs. CG; PE vs. CG) at each time point with respect to the baseline. The results show significant differences in most pre-post comparisons with medium and high effect sizes. Specifically, the total PSQ score for MM vs. PE was (t = -2.26, p=.02, d = .59) and for PE vs. CG was (t = 3.72, p<.01, d = .94). In the case of the subscales, it gave the following significant results: harassment-social acceptance for MM vs. PE (t = -2.59, p=.01, d = .68) and for PE vs. CG (t = 4.15, p<.01, d = 1.05); overload for MM vs. CG (t = 2.36, p=.02, d = .60) and for PE vs. CG (t = 3.07, p<.01, d = .78); and irritability-tension-fatigue for MM vs. PE (t = -2.06, p=.04, d = .54) and for PE vs. CG (t = 3.20, p<.01, d = .81).
Finally, we have evaluated the intragroup efficacy of MM and PE interventions derived from the significant major effects of the time factor. As seen in Table 4, the set of analysed variables showed significant differences (all <.01), with effect sizes ranging from moderate to high for MM in GHQ (total score) and coping dimension, and in the pre-Fup1 comparisons for self-esteem and stress dimensions.
Point 4.c. In Table 3 a value is missing in the MM v. CG comparison of irritability. Please clarify.
Response 4c: The value is .26. We have corrected the error on Table 3. Thank you for your observation.
Point 5.a There are two “Figure 1” figures in your article. The second Figure 1 uses a different acronym legend, “MF” presumably for “MM.” Please clarify.
Response 5.a.: We have corrected the error on Figure 2. Thank you for your observation.
Point 5.b. Please include standard deviation error bars on these graphs in order for the reader to be able to interpret them, and consider adding notations for significantly different comparisons.
Response 5.b.: In addition to replacing MF with MM, we have changed the graphics to include the standard deviation error bars. As you can see, these overlap and make their interpretation difficult. We'll leave it up to the reviewer's discretion. (PLEASE SEE THE ATTACHMENT)
Point 6. Your Discussion asserts that MM improves mental health better than PE, but I disagree and see no data that consistently supports that interpretation. Please clarify, and reference the supporting data.
Response 6: Line 355 states 'However, in general terms, no great differences have been observed between the results of MM and PE regarding mental health', since, in comparison, there are no significant differences, as can be seen from the interaction factor shown in Table 2 for GHQ. We then state that 'Although no significant differences have been found when comparing MM and PE, some interesting differences have been observed within some of the specific dimensions of the questionnaires.' Therefore, we suggest including Table 4, which was previously in the supplementary material, to improve reading comprehension.
Nonetheless, line 361 certainly generated confusion. Therefore, and in response to his observation, we believe that the following sentence would be better worded: 'Thus, MM seems to have an impact on the overall improvement of mental health and especially...' We hope that with this change everything will be better clarified.
Point 7.a. The limitations of your study are not sufficiently addressed. How was adherence to the intervention regimen controlled or monitored?
Response 7a: (line 216) Adherence to the intervention was controlled through registration sheets where each participant recorded each day the daily practice time, both during the intervention and after it, during the 6 subsequent months. With these records, the monitors of each group controlled weekly adherence to the practice. Those participants who did not complete a minimum level of practice (70%) were excluded from the experiment.
To improve understanding, we will replace the phrase 'All participants kept a daily record of their practice.' (line 216), with the phrase 'All participants kept a daily record of their practice in order to control that their adherence to the practice was at least 70%'.
In addition, with this information, we conducted a correlation study between the intensity of the practice (number of exact days) with the data in SPQ, GHQ and IGA. It corresponds to Table 4 of the supplementary material. There were no significant results.
Point 7.b. The waitlist control group appeared to trend toward improvement in some measures; did the control group participate in weekly lunchtime interactions of a controlled nature? Was there evidence that they were taking part in an intervention? Please discuss what a proper control group would look like and how closely your group matches that ideal. You mention that your intervention groups continued their interventions after the study treatment period. Is it possible the control group members created their own intervention while “waiting?”
Response 7.b.: Thank you for your observation. We too are surprised, although the results were not significant in any case. Thanks to the daily records and feedback given to the human resources department that recruited them, we know that the control group continued to take their lunch breaks as usual (eating with colleagues, taking the time to deal with personal matters, or catching up with work). They did know that they were taking part in an experiment about food breaks, since they were asked to record their usual activity during the meal, give out saliva samples for collection, and fill out the questionnaires. Aside from that, we believe that they were unaware of the specific objective of the experiment, beyond what they would have been able to deduce from the items in the questionnaires. However, we have not been able to control, whether there has been any 'leakage' of information from the intervention groups, such as, for example, on the technique of mindfulness. In any case, we consider that those would have been isolated 'cafeteria conversations' with little effect. We believe that this could not explain the results of the overall control group.
Nevertheless, due to his observation, we have included on line 351 the following expression: 'Last, we cannot find an explanation for the slight improvement in the results of the control group on perceived stress. However, these are not significant results'.
Point 8. Based on the data presented, I cannot agree that your conclusion that MM improving global mental health is supported by your results. Please clarify in your discussion what data supports this conclusion.
Response 8: In order not to overload the manuscript with tables, the results to which we refer were presented in the supplementary material, in Table 4. However, after their assessment, we consider it desirable and necessary that this table be in the main text.
It shows that the results of the simple effects of the MM time factor are significant in the totals of the GHQ score, of the MM coping subscale in all comparisons, and of the self-esteem and stress subscales in pre-Fup1.
We hope that the inclusion of Table 4 in the manuscript will aid in the understanding of the data.
